# Variability of Functional Groups of Rhizosphere Fungi of Norway Spruce (*Picea abies* (L.) H.Karst.) in the Boreal Range: The Wigry National Park, Poland

**DOI:** 10.3390/ijms241612628

**Published:** 2023-08-10

**Authors:** Jolanta Behnke-Borowczyk, Robert Korzeniewicz, Adrian Łukowski, Marlena Baranowska, Radosław Jagiełło, Bartosz Bułaj, Maria Hauke-Kowalska, Janusz Szmyt, Jerzy M. Behnke, Piotr Robakowski, Wojciech Kowalkowski

**Affiliations:** 1Faculty of Forestry and Wood Technology, Poznan University of Life Sciences, Wojska Polskiego 71C, 60-625 Poznań, Poland; robert.korzeniewicz@up.poznan.pl (R.K.); adrian.lukowski@up.poznan.pl (A.Ł.); marlena.baranowska@up.poznan.pl (M.B.); radoslaw.jagiello@up.poznan.pl (R.J.); bartosz.bulaj@up.poznan.pl (B.B.); maria.hauke@up.poznan.pl (M.H.-K.); janusz.szmyt@up.poznan.pl (J.S.); piotr.robakowski@up.poznan.pl (P.R.); wojciech.kowalkowski@up.poznan.pl (W.K.); 2School of Life Sciences, University Park Nottingham, Nottingham NG7 2RD, UK; jerzy.behnke@nottingham.ac.uk

**Keywords:** fungal diversity, mycorrhizal fungi, natural regeneration, trophic groups

## Abstract

Rhizosphere microbial communities can influence plant growth and development. Natural regeneration processes take place in the tree stands of protected areas, which makes it possible to observe the natural changes taking place in the rhizosphere along with the development of the plants. This study aimed to determine the diversity (taxonomic and functional) of the rhizosphere fungal communities of Norway spruce growing in one of four developmental stages. Our research was based on the ITS region using Illumina system sequencing. Saprotrophs dominated in the studied rhizospheres, but their percentage share decreased with the age of the development group (for 51.91 from 43.13%). However, in the case of mycorrhizal fungi, an opposite trend was observed (16.96–26.75%). The most numerous genera were: saprotrophic *Aspergillus* (2.54–3.83%), *Penicillium* (6.47–12.86%), *Pyrenochaeta* (1.39–11.78%), pathogenic *Curvularia* (0.53–4.39%), and mycorrhizal *Cortinarius* (1.80–5.46%), *Pseudotomentella* (2.94–5.64%) and *Tomentella* (4.54–15.94%). The species composition of rhizosphere fungal communities was favorable for the regeneration of natural spruce and the development of multi-generational Norway spruce stands. The ratio of the abundance of saprotrophic and mycorrhizal fungi to the abundance of pathogens was high and promising for the durability of the large proportion of spruce in the Wigry National Park and for forest ecosystems in general.

## 1. Introduction

The rhizosphere is where interactions between microorganisms and plants take place. Roots in the rhizosphere have an intimate association with the microbes that inhabit the soil [1]. This rhizosphere microbiome includes microorganisms associated with the surface of the roots and the soil layer adjacent to them [2]. For forest ecosystems, fungi are of particular importance, as they have a comprehensive effect on the operation of both the tree and the entire woodland community [3]. Changes in taxonomic and functional diversity, and the abundance of fungal communities, depend on environmental conditions and the health, vitality, and developmental stages of host plants [4,5,6,7,8]. Many plant-associated microorganisms affect seed germination, seedling vigour, growth and development, nutrition, disease susceptibility, and plant productivity [9]. The community consists of species that perform several functions in the soil, ranging from saprotrophs through endotrophs or mycorrhizal fungi to pathogens. Each functional group affects, directly and indirectly, the dynamics of the forest ecosystems.

Ectomycorrhizal fungi combine with plant roots in symbiotic systems that provide plants with water and mineral salts and oxidize organic matter to acquire nitrogen, dissolve phosphate, and reduce pathogens [10]. The vast majority of mycorrhizal fungi are generalists. Once a fungus has colonized a host plant, its mycelium can grow and extend for a long distance in the soil and reach and colonize the roots of neighboring plants of the same or different species [11]. Mycorrhizal fungi connect individual plants to form common mycorrhizal networks (CMN) [12,13,14]. This can modulate the survival and behaviour of connected plants, their competitiveness and cooperation (symbiosis), and consequently affect plant diversity [15,16]. For example, improvement in seedling survival has been reported due to plant-fungal interactions [17,18]. Saprotrophic fungi break down soil organic matter to release carbon, thereby contributing to the nutrient cycle [19]. Elements used to build new cells are released as a result of their action. Endophytes colonize plants’ tissues without visible disease symptoms. They inhabit plants by vertical transfer (from seeds) or penetrate from the external environments [3,6]. Suppression of diseases by beneficial microorganisms in the rhizosphere can occur using several mechanisms of action, such as antagonism associated with the production of antibiotics, antifungal metabolites, competition for space and nutrients with phytopathogens and other microorganisms harmful to the rhizosphere, and induction of resistance in plants. The pathogenic fungi cause plant diseases, impairing growth and reproduction and they may even cause the death of host plants. They require conditions conducive to infection, such as drought or wounds [20].

The rhizosphere microbiome changes and adapts to current environmental conditions, i.e., biotic and abiotic stress [21,22,23]. In the early stages of plant growth, when there is a poor root density, the fungal community is dominated by species with relatively long hyphae and fast growth (e.g., the genera *Cenococcum*, *Cortinarius*, *Inocybe*, *Laccaria*, *Scleroderma*, *Sebacina*, *Thelephora i Tomentella*, and the order Pezizales) [24]. As the plants in the community develop, the ectomycorrhizal species, in particular, become more diverse [24,25]. 

Following disturbances arising from storms, fires, or pest outbreaks, forest stands in protected areas such as national parks typically comprise some surviving large living trees, significant deadwood, gaps in the canopy, and coexistence of different developmental stages, and are then succeeded by natural regeneration dynamics. As a result, the newly established stands are diverse in terms of age and biometric characteristics [26]. Understanding the mechanisms and conditions that influence natural regeneration is a critical topic in forest ecology and management, especially in the context of the ongoing climate crisis [27]. In recent years, the role of microorganisms potentially involved in natural regeneration has attracted much attention. The development of molecular methods, including next-generation sequencing, has enabled the full spectrum of microorganisms from tested samples to be identified and quantified. Consequently, this progress in technology has changed the possibilities for research in this field, from just observing a single taxon to monitoring the whole community and facilitating the identification of the interactions between functional and phylogenetic groups [28].

Norway spruce (*Picea abies* L. (H.Karst.)) is one of the dominant species of coniferous trees in the northern temperate and boreal forests of Europe. It grows in various climatic and edaphic conditions, but it is best adapted to cooler climates. It most often occurs on acidic and nutrient-rich soils with good moisture availability [29,30]. There are two distributional domains of this species in Poland—eastern (boreal) and southern (lowland and mountainous). We are increasingly observing the weakening of the health of spruce stands throughout Europe due to a variety of abiotic and biotic factors, which are reflected in the mass die-off of populations [31]. Therefore, it is crucially important to identify the factors that could promote the natural regeneration and development, and enhance the resilience of this species in its natural environment. 

The current study aimed to determine the relationship between the developmental stages of spruce and the diversity of rhizosphere fungal communities, with a focus on the function performed by individual taxa. The following hypotheses were formulated: (i) the species composition and abundance of particular functional groups of rhizosphere fungal communities will differ between the development stages of Norway spruce, from the seedling to the mature old trees in the forest; (ii) along with the change in the developmental stages, there will be a concurrent change in the share of saprotrophs and mycorrhizal fungi, and (iii) a high abundance of saprotrophs and mycorrhizal fungi will be associated with low abundance of pathogenic fungi.

## 2. Results

Following normalization, a library of 412,560 Operational Taxonomic Units (OTUs) was obtained, representing 599 taxa. The identified taxa belonged to 6 phyla: Ascomycota, Basidiomycota, Chytridiomycota, Entomophthoromycota, Mucoromycota, and Zoopagomycota. In the four stages of development, the range of individual taxa ranged from 32.84 to 55.88% for Ascomycota and Basidiomycota from 31.38 to 67.03%. Figure 1 shows a clear trend reflecting a decreasing share of Ascomycota taxa and an increasing share of those from Basidiomycota with consecutive stages in tree development. Statistical tests revealed a significant difference between these two phyla (Table 1), but no substantial differences were found between tree stands. 

Out of 339 Ascomycota taxa, we identified 298 in the rhizosphere of the seedling, 278 in small saplings, 289 in large sampling, and 282 in mature trees, and of these 243 were common in all stages of development. For Basidiomycota, 424 taxa were identified, of which 397 were in the rhizosphere of the seedings, 367 in small sampling, 390 in large sampling group, and 370 in mature trees, and 323 were common in all stages. The average number of taxa in the rhizosphere of the seedling was 783, in small saplings 726, in large saplings 765, and in mature trees 739, including 639 common taxa for all stages (Figure 2). The number of taxa found only in seedlings was 15. Five were found only in saplings up to 50 cm, nine in saplings above 50 cm, and 9 in mature trees (Figure 2). 

It was observed that the abundance of genera, such as *Amphinema* (1.20–0.18%), *Tylospora* (0.34–0.006%), *Thelephora* (0.87–0.03%), (belonging to mycorrhizal), *Armillaria* (0.50–0.13%), *Bryochiton* (4.14–0.02%), *Coniothyrium* (1.96–0.20%), *Drechslera* (1.51–0.02%), *Heterobasidion* (1.13–0.52%), *Scolecobasidium* (0.21–0.00%), *Zeloasperisporium* (0.15–0.00%) (belonging to pathogens), *Archaerhizomyces* (0.03–0.49%), *Basidiobolus* (0.24–0.04%), *Entoloma* (1.70–0.47%), *Exophiala* (2.43–0.57%), *Guttulispora* (0.70–0.01%), *Ischonoderma* (0.70–0.09%), *Kavina* (1.13–0.09%), *Kondoa* (0.54–0.04%), *Mycena* (1.26–0.45%), *Pyrenochaeta* (11.78–1.39%), *Stereum* (1.20–0.18%), *Thysanophora* (0.36–0.06%), *Tulasnella* (1.57–0.29%), (belonging to saprotrophs) was highest in the seedlings or small saplings. The Amanita (0.33–0.04%), *Inocybe* (2.59–0.78%), *Hygrophorus* (0.56–0.00%), *Paxillus* (0.43–0.04%), *Sceroderma* (0.15–0.04%), *Tomentella* (15.94–4.54%) (belonging to mycorrhizal), *Diplodia* (2.59–1.57%), *Foliporia* (1.04–0.33%) (belonging to pathogens), *Aureobasidium* (0.36–0.08%), *Cyanosporus* (0.22–0.01%), *Fibroporia* (0.24–0.01%), *Lycoperdon* (2.11–0.13%), *Sclerogaster* (1.87–0.15%) (belonging to saprotrophs) was highest in large saplings or mature tree (Figure 3). 

An accurate identification up to the species or genus level was conducted to enablae the functional characteristics of taxa to be grouped, as most functional elements of fungi are conserved at the generic level and sometimes at higher taxonomic levels [32,33,34]. Analyzing the communities with respect to functional groups, saprotrophs were the most numerous group. Across four development stages, the average abundance of individual functional groups varied and in seedling, we noted 51.90% OTUs, in small saplings 44.02%, in large saplings 40.29%, and in mature trees 43.12% A decreasing trend in the proportion of saprotrophic fungi was observed in small saplings and large saplings as development progressed from seedlings to mature trees. In the case of mycorrhizal fungi, the average of OTUs in the rhizosphere of seedlings was 16.97%, saplings up to 50 cm (small saplings) 20.35% saplings above 50 cm (large saplings) 28.45%, and mature trees (mature trees) 26.76%. For pathogens, the average relative abundance in the rhizosphere in seedling ranged from 14.77% OTUs, in small saplings from 16.70% in large saplings from 6.81%, and in mature trees from 11.57% (Figure 4). Similarly, the developmental stage of trees was an important factor in explaining the variability of the abundance of functional groups: ectomycorrhizas and pathogens (Table 2). Figure 3a shows a general trend reflecting an increasing share by the former, and Figure 3b a falling trend in the latter, with consecutive three developmental stages. The abundance of saprotrophs, which dominate quantitatively, showed a decreasing trend from the seedling stage up to the mature tree stage (Figure 3c), which was just marginally the wrong side of our cut-off for significance (*p* ~ 0.06 level).

As a result of the alpha diversity analysis, communities differing in terms of species composition and abundance of individual taxa were identified. However, statistical significance was only found in the post-hoc test of the ChaoI index. The highest mean value of Chao was for small saplings (Table 3). Non-significant differences were found within fixed factors for the Shannon and Simpson indices.

## 3. Discussion

In this paper, we have identified the rhizosphere fungal taxa of Norway spruce from the Wigry National Park, Northeastern Poland, and determined their functional groups, on four developmental stages of these conifers: on current year seedlings, saplings up to 50 cm (small saplings) and over 50 cm in height (large saplings) and mature trees. As expected, the greatest alpha diversity was in the rhizosphere of small spruces. The highest abundance of Ascomycota was in the rhizospheres of current-year seedlings and small saplings up to 50 cm in height, while the highest abundance of Basidiomycota was in the rhizosphere of saplings that were over 50 cm in height and in mature trees. The Ascomycota phylum dominates most ecosystems. This is related to their ecological requirements and functional features, which translate into the abundance of spores and how they spread [35]. The ability to disperse is the main factor determining the composition of early succession fungal communities. In the case of the Ascomycota, saprotrophic taxa predominate [36], and therefore, as expected the share of saprotrophic species in our study was higher in the rhizospheres of younger trees compared with older trees. The more significant share of saprotrophic fungi in the rhizosphere of the juvenile development stages of spruce stands can be explained to some extent by the appearance of natural regeneration in the gaps in the tree canopy formed after the death of trees, next to decaying pieces of wood and even on rotten and only partially decomposed spruce trunks [37], and a relatively small root tissue volume in the sample. The abundance of wood lying in gaps in the forest canopy is conducive to the mass occurrence of saprotrophs, and at the same time, these spaces create favorable conditions for the emergence and development of a young generation of trees. Among the saprotrophic fungi, genera such as *Aspergillus*, *Talaromyces*, *Phlebia*, *Phlebiopsis*, and *Penicillium* prevailed in the Wigry National Park. These taxa are considered antagonists to the genera *Armillaria* and *Heterobasidion* [38,39,40]. The mycelium and its metabolites reduce the extent of rot caused by *Armillaria* spp. and the extent of base necrosis caused by *Heterobasidion* in Scots pine and other tree species. These taxa compete with pathogens for living space and food. They have the ability to dissolve phosphates and produce antifungal agents, e.g., citrinin, griseofulvin, fellutanin A, and archidic acid [41]. 

The presence of the genus *Mycena* in the analyzed community indicates that decaying wood (like that of spruce roots) lies near sampling sites. *Mycena* spp. is the cause of white rot decay and has been previously identified as a significant component of fungal communities in later stages of root [35] or wood decay [36]. Our results did not confirm the abundant occurrence of this taxon in the late developmental stages of trees, which may be due to the different (rhizosphere not soil) material collected for the current research project. The rhizosphere consists of small roots and the soil adjacent to it, and not of decomposing plant tissues.

It is known that saprotrophs and mycorrhizal fungi are essential in creating suitable conditions for future stands of trees to grow and develop from the juvenile stage [37,38,39]. As these grow, the diversity and abundance of mycorrhizal species increase. Among the mycorrhizal fungi, the most numerous were: *Cortinarius*, *Amanita*, *Inocybe*, *Pseudotomentella,* and *Tomentella*. Species of these genera are common mycorrhizal species found in Pinaceae worldwide [40,41]. Taxa from the genera *Cenococcum*, *Cortinarius*, *Inocybe,* and *Tomentella*, as well as the order Pezizales in the above work, were identified from the early stages of development [40], which confirms their importance as the first colonizers of roots. Some of these mycorrhizal fungi have features of saprotrophs and usually show a poor ability to decompose carbon compounds [10]. Fungal communities make it possible for trees to adapt to the current environment [32,42], an ability associated especially with the genus *Amanita*, among others. This genus is a good indicator of the presence of mature trees. After the removal of the host plant, it disappears very quickly from the site [35]. In our research, this genus was most abundant in the rhizosphere of mature trees. Representatives of this genus were scarce in the rhizosphere of the other developmental stages. *Thelephora* is an ectomycorrhizal basidiomycetes genus, of which all constituent species. form ectomycorrhizal relationships with diverse plants [39]. As mycorrhiza-formers, *Thelephora* plays a very important role in pioneering microhabitats in coniferous forests and is also involved in the decomposition of dead wood [43]. *Thelephora wakefieldiae* is one of the most important fungi establishing symbiotic relationships with Norway spruce in the Wigierski National Park. It dominated in each of the samples, although it was most common in large saplings.

We observed the prevailing dominance of saprotrophs and mycorrhizal fungi in our study sites. Accordingly, the high abundance of mycorrhizal species constitutes a natural biological defense against pathogens, especially those causing diseases of root systems, including fungi belonging to the genera *Fusarium, Verticillium*, *Typhula*, *Gaeumannomyces*, *Calonectria*, *Nectria*, *Rhizoctonia* and fungus-like organisms (Oomycota): *Phytium* and *Phytophthora* genera, as well as plant-parasitic nematodes. These species are responsible for the most significant threat to the early stages of development of saplings; among other pathogenic consequences, they cause the blight of seedlings. In mature tree stands, mycorrhizal symbiosis is common and has a positive effect on the growth and development of trees.

We identified also taxa belonging to the genus *Alternaria*, which includes species of opportunistic plant pathogens contributing to losses in fruit and seed storage, found worldwide [44]. *Alternaria* spp. cause 20–80% agricultural losses in field, horticultural, plantation, and forest plants, as well as post-harvest storage. The *Alternaria* species cause needle blight in *Pinus bungeana* in China [45]. Infected seeds suffer from blight and death of the necrotic cotyledons. Infection occurs through the lower leaves of seedlings, which are infected by spores from the soil. Initially, there are slight changes on the leaves, which develop to a size of 5–15 mm in diameter with light and dark concentric rings. The spores are carried also by rain and wind and infect other tissues. Wet and dry weather favors the life cycle (germination and spore release) of *Alternaria* [46].

The genus *Diplodia* was identified in the analyzed community. *Diplodia sapinea*, a species common throughout Europe, although in our research, it was rarely cause dieback of shoots of coniferous trees [47,48,49]. 

Among the pathogens, we identified taxa responsible for diseases of the root system belonging to the genus *Heterobasidion*, including *H. parviporum* and *H. annosum,* and a small contribution of *Armillaria* spp. *Heterobasidion parviporum* mainly infects spruce [50], while *H. annosum* has a broader host spectrum and infects both Scots pine and Norway spruce [51]. The highest abundance was in seedlings, although we did not observe any pathological symptoms of *Heterobasidion* infection. This pathogen quickly kills young trees, while in older trees, it causes long-term disease, consisting of the decay of wood and roots of the lower parts of the trunk. In older spruces decomposition only occurs in the heartwood, and as a result, their vital functions remain normal [52]. These fungi are not present in smaller roots.

The genus *Armillaria* causes dieback of trees older than 2 years. Colonization of the root system of large and old trees is slow and sometimes takes decades. In younger trees, the dying process is rapid and lasts from several months to a year after infection. The presence of the rhizomorph *Armillaria* spp. was found in the soil in the stand in Mikołajewo, while in the analyzed community the abundance of the DNA of this pathogen was low. However, the abundance of pathogens and mycorrhizal species did not differ significantly between the plots. Statistical differences occurred in the case of saprotrophs for this area. In the case of *Armillaria* spp., the risk of infection increases as a result of drought. Tree roots weakened by drought are more susceptible to infection with rhizomorphs of the pathogen [53]. However, it should be remembered that *Armillaria* has so far been identified in roots larger than 2–3 cm, while in this study we investigated only the fine root.

*Pyrenochaeta* inhabit soil and plant debris worldwide and are well-known as pathogens of various plants. Several species of *Pyrenochaeta* fungi infect plants, particularly plant roots [54]. In the environment, numerous *Pyrenochaeta* species are found as saprotrophs in soil, plant debris, and wood [55]. Some species have been identified as tree endophytes. Some of the *Pyrenochaeta* species cause serious plant diseases in agriculture and forestry [56]. *Herpotrichia juniperi* and *Neopeckia coulteri* (*Pyrenochaeta* anamorphs) cause shoot and needle diseases of conifers [57]. For example, *Pyrenochaeta parasitica*/*Nematostoma parasiticum* occurs on *Picea* and *Tsuga* (the Herpotrichia needle browning) [58]. The species has been recorded in Austria, Switzerland, Denmark, Germany, Norway, Great Britain, and Poland [57,59], and sporadically in North America [60]. In our results, we identified this taxon only at the genus level. Fungi belonging to this genus may reduce the young spruce generations in the Wigry National Park.

Our study provides new insights into the variation of fungal communities among the developmental stages of spruce stands, however, we are aware of the limitations of our findings. Even within the same genus, some species may be symbiotic and others pathogenic. There are controversies in concluding that a species is pathogenic based on its ITS sequence alone. Even for species that are clearly identified as plant pathogens, there are reports of complex symbiotic relationships in which they do not express pathogenicity but rather enhance the plant’s immune system [61].

The fungal community consisting of a large abundance of saprotrophs and mycorrhizal species significantly reduces pathogen threats to the Norway spruce. In addition, plants have access to the elements necessary for life and for building new cells Therefore, the fungal community identified in Wigry National Park potentially benefits the natural regeneration, growth, and development of Norway spruce.

## 4. Materials and Methods

### 4.1. Sampling Site Description

Three uneven-aged stands with a differentiated vertical and species structure, and a significant share of the Norway spruce were selected for the study (Table 4). The *Serraulo-Pinetum* (W.Mat. 1981) J.Mat. 1988 (subboreal mixed forest) community dominates all stands. They were chosen to represent three parts of the Park—north, south-western, and south-eastern, where forests dominate. Since 1989 these stands have been excluded from silvicultural management. They are protected from interference and are reserved for scientific study, including observation of natural processes as a main objective (Poland, Regulation of the Council of Ministers of 27 June 1988—Journal of Laws No. 25 of 21 July 1988, item 173).

### 4.2. Sampling

The collection of research material was carried out in June 2022. For the analysis of the communities, twenty trees were selected randomly in each stand as follows. Five vigorous individuals of Norway spruce, with no disease symptoms, from four forest strata were selected in each stand: current-year seedlings with visible cotyledons from the undergrowth, saplings up to 50 cm in height, and over 50 cm in height from the small and large saplings strata respectively, and mature trees with visible generative organs from the overstorey. Their roots were tracked, and surrounding soil at a depth of up to 15 cm was collected. Finally, sixty samples were collected (three stands × four forest strata × five samples = 60). Samples were placed in a plastic container and immediately taken to the laboratory. Comparably sized subsamples of fine root branches (2 cm long with soil) were placed in 2 mL screw-cap tubes. Samples were stored at −20 °C until molecular analyses.

### 4.3. DNA Extraction and PCR Amplification

The collected material was ground in a mortar chilled to −70 °C. Then DNA was isolated using the DNeasy PowerSoil Kit (QIAGEN, Hilden, Germany). The DNA was then purified using the Anti-Inhibitor Kit (A&A Biotechnology, Gdynia, Poland). Total DNA was sent for sequencing and preparation of OTU (operational taxonomic unit) libraries at Genomed SA (Warsaw, Poland). Fungal community analysis was performed based on the Internal transcribed spacer 1 (ITS1) region using specific primers ITS1FI2 5′-GAA CCW GCG GAR GGA TCA-3′ [62,63] and 5.8S 5′-CGC TGC GTT CTT CAT CG-3′ [64]. PCR was performed using Q5 Hot Start High-Fidelity 2X Master Mix (New England Biolabs, Ipswich, MA, USA). Reaction conditions were as recommended by the manufacturer. Sequencing was performed on a MiSeq sequencer in paired-end (PE) technology. Negative samples (without DNA) were also sequenced to remove artifacts.

### 4.4. Sequence Processing and Statistical Analysis

The obtained sequences were subjected to bioinformatics analysis to generate the OTU library. The first step was classifying reads to species level using the QIIME2 software package (ver. 2017.6.0). The study consisted in removing the adapter sequences in the cutadapt software (v4.4) and then in the qualitative analysis of the reads. Low-quality sequences (quality < 25, length up to 30 bp) were removed. In the next group, paired sequences were combined (fastq-join algorithm) [65], chimeric sequences were removed (usearch algorithm), clustering was performed based on the selected reference sequence database (uclust algorithm) and taxonomy was assigned based on the NCBI (National Center for Biotechnology Information, https://www.ncbi.nlm.nih.gov/ (blast algorithm ver. BLAST+ 2.14.0)) (accessed on 27 May 2023) [66]. For identification, the percentage of sequence similarity with the reference sequence was assumed to be 98–100%, with a minimum coverage of 90% [67,68]. Taxa were identified down to the lowest possible taxonomic level. OTU sequences not belonging to Fungi or Oomycota were removed from further analysis. Further, OTUs were assigned to ecophysiological categories using FungalTraits [69], with subsequent manual corrections. Finally, the taxa were divided into pathogens, saprotrophs, endotrophs, mycorrhizal fungi, and lichens. The applicable nomenclature was adopted from MycoBank (https://www.mycobank.org/, accessed on 13 May 2023). Rarefaction curves were determined for the obtained OTU library. Rarification samples were made to normalise the samples to one size. As a result of normalization, two negative samples were removed. The ChaoI species diversity index and Shannon diversity index and Simpson index were calculated using function richness (vegan) [70]. VennDiagram (https://bioinfogp.cnb.csic.es/tools/venny/, accessed on 14 May 2023) has been used to illustrate dependencies between number of taxa in different tree development stages.

### 4.5. Statistical Analyses

A two-way analysis of variance model was used to compare fungal frequencies expressed in OTU units from different tree stages (seedling, small samplings, large saplings, and mature trees) and stands (Lipniak, Mikołajewo and Powały). This model was applied to analyse six dependent variables: Ascomycota: Basidiomycota ratio, fungi functional groups (ectomycorrhizas, pathogens, and saprotrophs), and two alpha diversity indexes (Chao, Shannon and Simpson). From a total of 58 independent observations a few outlying data points were removed for the purposes of the individual analyses. This information is emphasised in the tables containing the ANOVA results. Homogeneity of variance within fixed factors (tree groups, stands) was checked by the Levene F test using the function leveneTest(car ver. 3.1-1). The goodness of fit of the data and residuals of ANOVA to a normal distribution was checked by Shapiro-Wilk test using the function shaprio.test (stats ver. 4.2.2). To meet this requirement of parametric models two variables were transformed (by natural logarithm or square root), as stated in the tables providing the results of analysis by ANOVA. For statistically significant factors, least square means were calculated (function emmeans(emmeasn ver. 1.8.2)) and the Tukey HSD post-hoc test was implemented to identify a significant difference between factor levels (function contrast(emmeans ver. 1.8.2)). All statistical analyses were performed in R 4.2.2 [71].

## 5. Conclusions

The gradual decrease in abundance of saprotrophic species and increase of ectomycorrhizal in consecutive tree developmental stages indicates favorable conditions for spruce stand regeneration. The communities were dominated by saprotrophic and mycorrhizal species, with a small share of pathogenic species. The many species antagonistic to *Heterobsasidion* spp. and *Armillaria* spp. are beneficial for the growth and development of multi-generation Norway spruce stands in the Wigry National Park. 

## Figures and Tables

**Figure 1 ijms-24-12628-f001:**
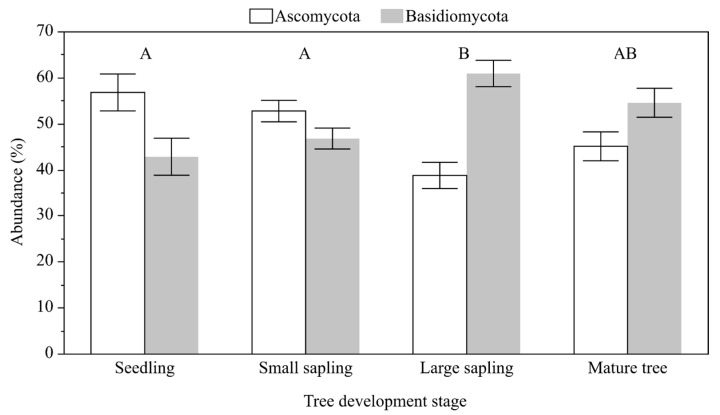
Mean and standard error of the mean for the relative abundance of taxa from the Ascomycota and Basidiomycota divisions in different tree developmental stages. The results of the Tukey HSD post hoc test are presented above the columns. Columns denoted by a different letter are significantly different (α = 0.05) for the Ascomycota/Basidiomycota. The results of the analysis of variance for the Ascomycota/Basidiomycota ratio are presented in Table 1.

**Figure 2 ijms-24-12628-f002:**
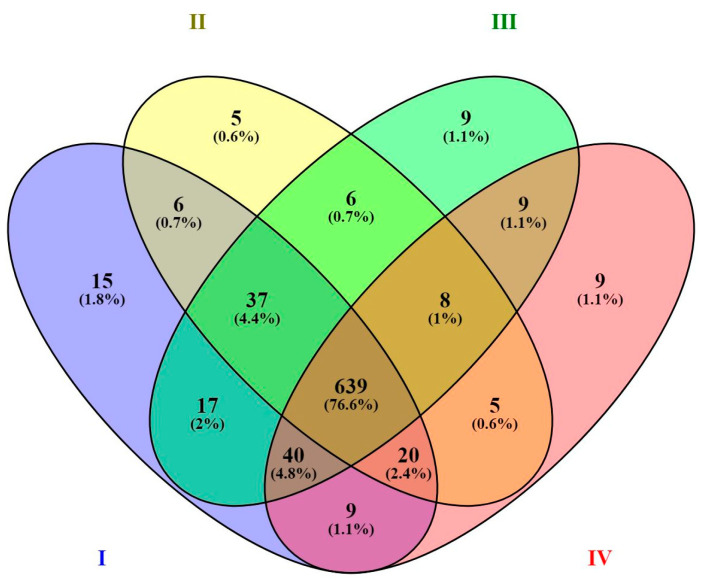
Venn diagram showing the number of taxa in different developmental stages. Treatments: (I) seedlings—current year plants with visible cotyledons; (II) small saplings up to 50 cm grown under the canopy of the mature stand, (III) large saplings over 50 cm in height, grown under the canopy of the mature stand, and (IV) mature trees.

**Figure 3 ijms-24-12628-f003:**
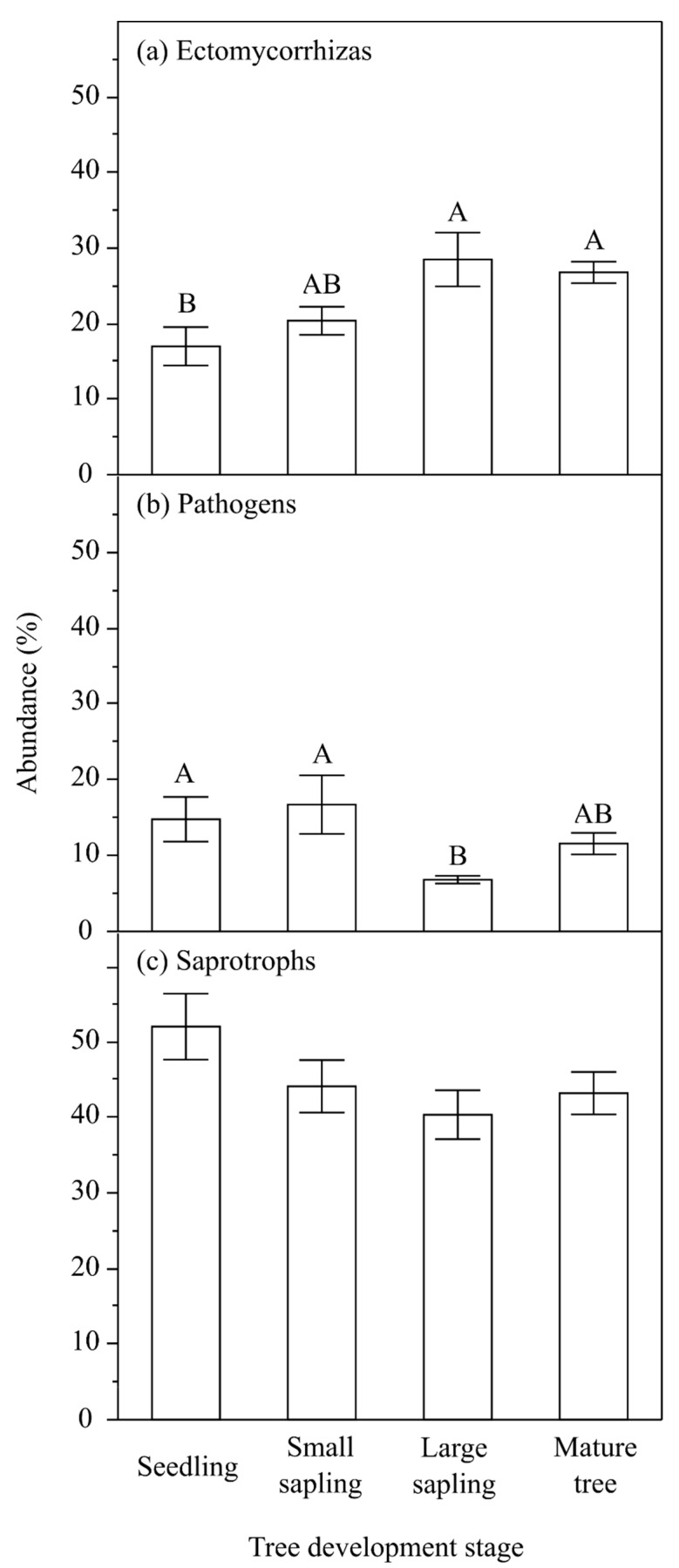
Mean and standard error of the mean for relative abundance of taxa classified into three functional groups: (**a**) ectomycorrhizas, (**b**) pathogens, and (**c**) saprotrophs. The results of the Tukey HSD post-hoc test are presented above the columns. Columns denoted by a different letter are significantly different (α = 0.05). The results of the analysis of variance are presented in Table 2.

**Figure 4 ijms-24-12628-f004:**
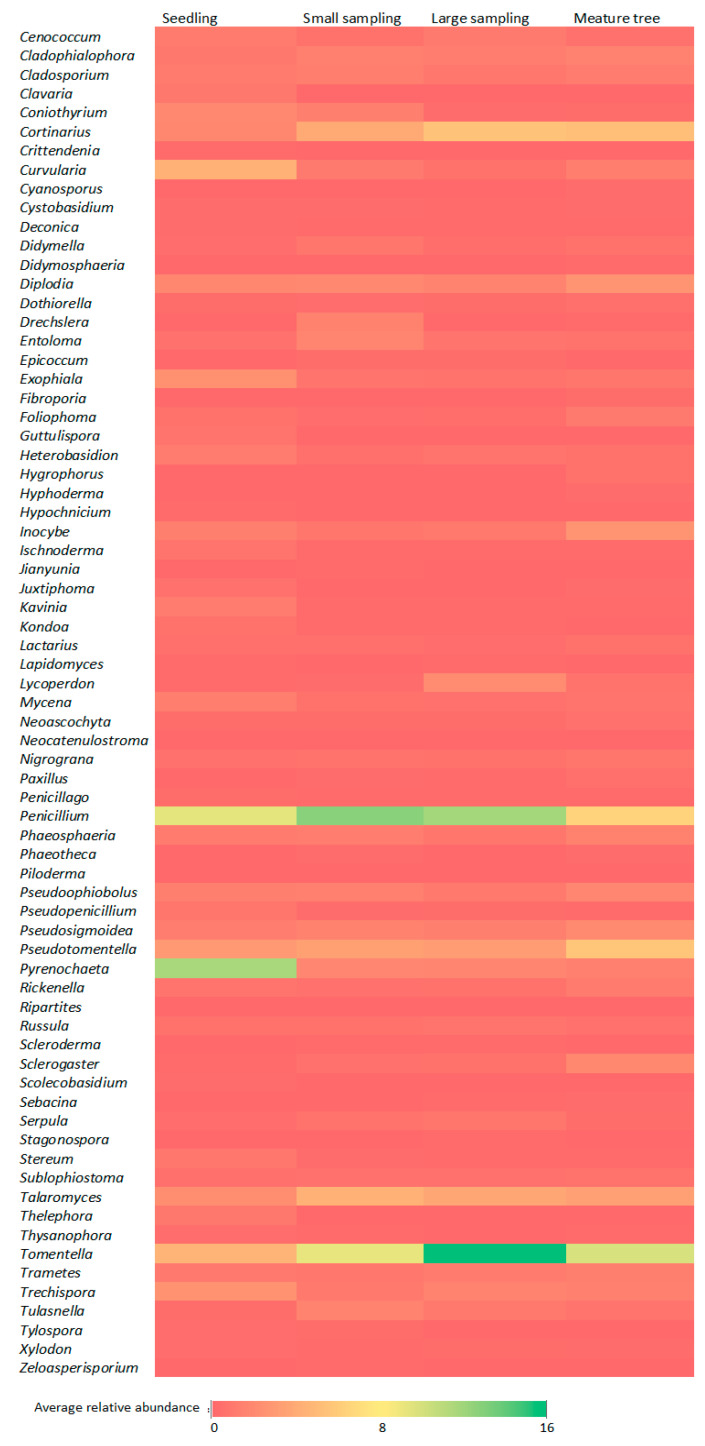
Heat map for the relative abundance of genera (the minimum threshold >1%).

**Table 1 ijms-24-12628-t001:** Results of the two-way analysis of variance for OTU Ascomycota: Basidiomycota ratio from different stands (S) and tree development stage (TDS) Df—degrees of freedom; F—the F-test value; P(α)—the probability of type I error.

Source of Variation	Df	F	P (α)
TDS	3	5.474	0.003
S	2	2.055	0.140
TDS × S	6	0.404	0.872

**Table 2 ijms-24-12628-t002:** Results of the analysis of variance for mean OTU of taxa classified to three functional groups (ectomycorrhizas, pathogens and saprotrophs) from different stands (S) and tree development stage (TDS). Df—degrees of freedom; F—the F-test value; P(α)—the probability of type I error.

Source of Variation	Df	Ectomycorrhizas		Pathogens		Saprotrophs	
		F	P (α)	F	P (α)	F	P (α)
TDS	3	5.616	0.002	3.942	0.014	2.598	0.063
S	2	1.363	0.266	0.984	0.382	5.522	0.007
TDS × S	6	1.538	0.187	1.766	0.127	1.792	0.122

**Table 3 ijms-24-12628-t003:** Mean, standard error of the mean and ANOVA results for the Chao1 and Shannon diversity indices from different tree development stages (TDS) and stands (S). Df—degrees of freedom; F—the F-test value; P(α)—the probability of type I error. Levels not connected by the same letter are significantly different (Tukey’s HSD test).

Tree Development Stage		Chao1		Shannon		Simpson	
		Mean (Standard Error)					
Seedlings		718 (61) b		3.91 (0.230)		0.986 (0.044)	
Small saplings		1213 (128) a		4.06 (0.209)		0.923 (0.023)	
Large saplings		845 (76) b		3.92 (0.195)		0.901 (0.036)	
Mature trees		891 (89) ab		4.23 (0.219)		0.932 (0.030)	
**Analysis of Variance**							
**Source of** **Variation**	**Df**	**F**	**P (α)**	**F**	**P (α)**	**F**	**P (α)**
TDS	3	5.307	0.003	2.249	0.097	0.270	0.847
S	2	1.296	0.283	2.477	0.096	0.122	0.886
TDS × S	6	1.046	0.408	2.126	0.071	2.012	0.083

**Table 4 ijms-24-12628-t004:** Vertical structure and species composition of the stands involved in the research. Ap—*Acer platanoides* L., Bp—*Betula pendula* Roth, Ca—*Corylus avellana* L., Ev—*Euonymus verrucosus* Scop, Fa—*Frangula alnus* Mill., Jc—*Juniperus communis* L., Lx—*Lonicera xylosteum* L., Pa—*Picea abies* (L.) H.Karst., Ps—*Pinus sylvestris* L., Pt – *Populus tremula* L., Qr—*Quercus robur* L., Sa—*Sorbus aucuparia* L., Tc—*Tilia cordata* Mill.; ns—not specified; * Data source: Forest Data Bank (https://www.bdl.lasy.gov.pl/portal/en), accessed on 30 May 2023. Data in the table are derived from the last update for the year 2013. The species are indicated as a gradient of decreasing share within the stratum.

Protection Zone, Compartment, GPS Location	Area [ha]	Forest Stratum	Cover [%]	Species Composition * [Age in Years]
Lipniak, 15f 54°08′53.4″ N 23°05′15.5″ E	9.75	Overstorey	80	Pa (116), Ps (116)
		II stratum	20	Pa (75), Pa (50), Bp (50), Tc (50)
		Large saplings	10	Pa (25)
		Small saplings	ns	Pa (5), Qr (3), Ap (3), Tc (3)
		Undergrowth	50	Lx, Ca, Fa, Sa, Ev
Mikołajewo, 210c 54°02′10.5″ N 23°08′55.0″ E	6.55	Overstorey	70	Pa (141), Ps (141), Pa (101)
		II stratum	30	Pa (76), Pa (46), Qr (46), Pt (46)
		III stratum	20	Pa (31), Qr (31)
		Large saplings	30	Pa (21), Pa (11), Qr (21)
		Small saplings	10	Pa (4), Qr (4)
		Undergrowth	30	Ca, Sa, Lx, Fa, Ev
Powały, 390c 53°59′59.7″ N 23°02′12.9″ E	7.98	Overstorey	60	Ps (121), Pa (85), Pa (121), Bp (121),
				Bp (85), Qr (85)
		II stratum	30	Pa (35), Pa (50), Bp (35), Bv (50),
				Qr (50), Jc (35)
		Large saplings	10	Pa (15), Bp (15), Qr (15)
		Small saplings	ns	Pa (3), Bp (3), Qr (3)
		Undergrowth	30	Ca, Sa

## Data Availability

Data supporting reported results can be found at https://doi.org/10.6084/m9.figshare.23733399.v1 (accessed on 23 July 2023).

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
