# Peer review of "Variability of Functional Groups of Rhizosphere Fungi of Norway Spruce (Picea abies (L.) H.Karst.) in the Boreal Range: The Wigry National Park, Poland"

_ijms, 2023, doi:10.3390/ijms241612628_

Round 1
Reviewer 1 Report
Please see attached file

Before submitting your manuscript, be sure to go through the article proofreading program.
Author Response
Reviewer 1
- Title. 1.1 Wouldn't it be better to show the scientific name in the title? for exampie: Norway spruce (Picea abies)
In the revised title, we show the scientific name of our study species.
- Abstract
2.1. The abstract should present the main findings of the study and should not include detailed experimental methods. For example: “....small saplings up to 50 cm, understorey saplings over 50 cm in height, and mature trees with crowns situated in the overstorey....”
The experimental methods have been removed from the Abstract.
2.2.“Saprotrophs dominated in the studied rhizospheres, but their share of the microbial communities decreased with the age of the development group”-> It's good to quantify the reduction (For example, the percentage decrease or the number of species.)
Changes in the percentage share of each group of fungi are given in lines 23-24.
2.3. List the keywords in alphabetical order.
Corrected. Picea abies has been removed from the list of key words because it is now shown it in the title.
2.4. Describe how the each diversity value varied.
This information is given between lines 23-27.
2.5 Provide the exact number (dominance ratio): The most numerous genera were: saprotrophic Aspergillus, Penicillium, Pyrenochaeta, pathogenic Curvularia, and mycorrhizal Cortinarius, Pseudotomentella and Tomentella
We have now provided information on the relative abundance of individual genera. However, we have not indicated the dominant genus, but only those whose abundance was the greatest in the sample (lines 25-27).
- Result and Discussion
3.1. As far as I know, microbial sequences obtained from cultured-independent methods should be submitted to the SRA database.We acknowledge that the SRA database is the most popular place for depositing sequences. However, we have decided to deposit our data in https://figshare.com/account/articles/23733399.
- In addition, the <Data Availability Statement> should state the following.
“Data Availability Statement: All the raw sequences obtained from this study were deposited at the National Center for Biotechnology Information (NCBI) Sequence Read Archive (SRA) under the project accession number 00000000.”
Check out and follow these two articles:
https://doi.org/10.3390/biology10020138
https://doi.org/10.1080/12298093.2020.1796413
We acknowledge that the SRA database is the most popular place for depositing sequences. However, we have decided to deposit our data in https://figshare.com/account/articles/23733399. Our reason for doing so is that, apart from fastq files, it is possible to deposit also other elements of our study in this database. Our deposition includes files of functional groups, and the rarefaction curve.
Data supporting reported results can be found at https://figshare.com/account/articles/23733399 (accessed on 23 July 2023).
3.3. Line 109-163: Wouldn't it bore readers to list the data that's already in the figure? Be brief and only state facts that are essential to the further discussion. If it's ambiguous to do so, why not combine the result and discussion?
3.4. Table 1, 2, 3, 4: Include only the title at the top of the table, and footnotes at the bottom of the table for any additional explanation. See the example below. Before submitting your manuscript, be sure to go through the article proofreading program.
For example:
Table 4. Chao 1 and the shannon diversity value.
* For the purposes of the Shannon diversity index analysis five outlying observations were excluded from the results. For Chao1 index results of Tukey HSD post-hoc test are indicated. Levels not connected by same letter are significantly different (α=0.05).
The titles and footnotes of the tables have been improved.
3.5. what is a, b, ab in table 4? what is df, f, p(a) in the script? what is HSD post-hoc test?
The statistical symbols are now explained in captions and footnotes of the figures and tables, and should be now self-explanatory.
3.6. Follow the formatting of the journal. Numbering of titles and subtitles are: 2., 2.1., 2.1.1.
The numbering of titles and subtitles have been amended accordingly
3.7. To correctly interpret Chao 1 richness and Shannon value variation between small saplings and the mature trees, it is necessary to apply the Simpson index. Chao 1 richness decreased, but Shannon diversity increased. This is likely due to changes in the dominance of certain fungal species, so it's worth checking the data on Simpson value. If certain fungal species became dominant in the mature trees, their dominance would have affected the stability of the soil fungal community. Also, perhaps the role of their dominance on the tree is important. Therefore, it is important to calculate the Simpson's index and see which fungal species have become dominant. Re-analyze the authors' data for the calculation of the Simpson value, and interpret it according to the derived Simpson value.
The Simpson index has been calculated and its variation among the factors and stands is compared. However, there is no significant signal from that variable.
3.8. Line 334: Study site-> sampling site description
Corrected. (line 362).
3.9. line 335-340: Is there any images of the sampling sites? or location on map? or the GPS location?
In the resubmitted, revised version of the manuscript, the geographical coordinates of the sampling sites are given (See table 1).
3.10. line 342: Material collection->sampling
Corrected (line 372).
3.11. line 354: Molecular analyses-> DNA extraction and PCR amplification
Corrected (line 384).
3.12. line 366: Bioinformatic analysis-> Sequence processing and statistical analysis
Corrected (line 396).
3.13. line 338-340: Which countries have implemented these policies? Include any relevant legislation or references.
The relevant reference is in lines 368-371.
3.14. line 336: what is ‘Serraulo-Pinetum’? You must write the scientific name. is it Pinus sibirica?
This is the name of a plant community following syntaxonomy. The authors of the taxonomic name and the English name of the community have been added.
.
3.15. line 336: What is the relationship between Picea abies and Pinus sibirica?
There is no mention of Pinus sibirica in the text.
3.16. figure 3. Some fungi were not shown. The top of the picture is cropped.
The figure shows only the types for which the abundance in at least one sample was > 1%. Since the work concerns functional groups and not specific identities of fungi, we concluded that it most meaningful to indicate only the most numerous types.
3.17. The conclusion part should summarize the new findings from the research. The paper should also discuss the limitations of the study and the future research directions based on the findings of the study. The abstract should also summarize the background and need for the study and the main findings. But the conclusion and the abstract don't match at all. Is the conclusion a significant finding? Or is it what is written in the abstract? I'm not sure what the main finding is that the author wants to show.
The main conclusion (lines 440446) is now in agreement with the conclusion in the Abstract (lines 29-32). We have also added a short paragraph about the limitations of our study in accordance with the reviewer’s advice (lines 348-354).
3.18. line 231-233: What is the rationale for the artificial division into functional groups (ectomycorrhizas, pathogens and saprotrophs)? Authors said that there are limitations of identification based on ITS region (Line 136-138). There are many species within a given fungal genus, and even within the same genus, some species may be symbiotic and others are pathogenic. Therefore, the species must be identified using further genes before it can be called a pathogen. There is too much controversy to be faced when a paper is published to determine a species as pathogenic based on its ITS sequence alone. In addition, even for species that are clearly identified as plant pathogens, there are reports of complex symbiotic relationships in which they do not express pathogenicity but rather enhance the plant immune system.
We are aware of the inconvenience of the ITS region and acknowledge that this is one of the limitations of our study. However, it is the region that is used conventionally for this type of analysis. Due to the disadvantages mentioned by the reviewer, we used a database that is based on genera. This Db uses information on trophic groups for a given genus - not species. It takes into account a primary lifestyle (a dominant role for a given type) and a secondary lifestyle. During the manual correction, we summed up both roles and took into account the conditions under which the sampling was conducted. If the database described the role of a genus as a saprotroph of wood and saprotroph of litter, we included it generally as a saprotroph. We are aware that, although described as neutral in communities (for example), nevertheless there are some taxa that may behave differently. Our analysis is based on best current knowledge and the long-standing experience of our research group.
3.19. line 287: Diplodia seriata was identified among the pathogenic species in the analyzed community. -> There is too much controversy to be faced when a paper is published to determine a species as pathogenic based on its ITS sequence alone.
We have amended this part of the manuscript. We have described the genera we identified and added information on those pathogens that we can be in forest. Line 307-312
3.20. line 226-134: “....higest in large saplings or mature tree”,“.... highest in the seeding or small saplings” -> Explain with exact numbers, or implement them in a diagram with exact numbers.
The exact numbers are now given (lines: 130-144 ). This information is shown on the heatmap (Fig 4.)
3.21. line 256-265: “This genus is an indicator for mature trees. After the removal of the host plant, it disappears very quickly from the site [42]. In our research, this genus was most abundant in the rhizosphere of mature trees. Representatives of this genus were scarce in the rhizosphere of the other developmental stages”,“..... It dominated in each of the samples, although it was most common in large samplings” ->Explain the exact numbers/ratios, or implement them in a diagram with numbers.
The exact numbers are now given (lines: 130-144 ). This information is shown on the heatmap (Fig 4.)
3.22. describe the full name of ITS region
Corrected. Line 390
Reviewer 2 Report
Paper's introduction provides a good background and clearly describes the study's goals.
Appropriate references are used to support the assertions.
Your figure 3 clearly illustrates the paragraph (lines 126 to 135) on the relative abundance of some selected OTUs but this concerns less than a hundred taxa..
I would have liked to have access to the study's total abundance table as additional data for the different stands and tree development stages.
Providing the rarefaction curves in supplementary data could also be interesting.
Lines 139 to 158: I believe you are describing figure 4 rather than figure 3.
Overall, the paper is well-constructed, the methodology is adapted to meet the objectives set, and the conclusions are supported by the results obtained and the references.
Author Response
Reviewer 2
Paper's introduction provides a good background and clearly describes the study's goals.
Appropriate references are used to support the assertions.
Your figure 3 clearly illustrates the paragraph (lines 126 to 135) on the relative abundance of some selected OTUs but this concerns less than a hundred taxa..
I would have liked to have access to the study's total abundance table as additional data for the different stands and tree development stages.
We provide the table with all data as files in https://doi.org/10.6084/m9.figshare.23733399.v1 .
Providing the rarefaction curves in supplementary data could also be interesting.
We have provided all of our data in https://doi.org/10.6084/m9.figshare.23733399.v1 . Inclusion of the refraction curves in the manuscript text would be a repetition of the results shown elsewhere.
Lines 139 to 158: I believe you are describing figure 4 rather than figure 3.
Yes, this was our mistake. The number of this figure has been corrected.
Overall, the paper is well-constructed, the methodology is adapted to meet the objectives set, and the conclusions are supported by the results obtained and the references.
Thank you very much for all the remarks and suggestions. We have corrected the manuscript accordingly.
Reviewer 3 Report
This manuscript presents an important and very interesting dataset, with appropriate conclusions. I foresee that these data will continue providing useful information to making management decisions well into the future.
Author Response
Reviewer 3 :
This manuscript presents an important and very interesting dataset, with appropriate conclusions. I foresee that these data will continue providing useful information to making management decisions well into the future.
We warmly thank for this opinion.
Round 2
Reviewer 1 Report
Please see attached file.
